# Cell Heterogeneity and Variability in Peripheral Nerve after Injury

**DOI:** 10.3390/ijms25063511

**Published:** 2024-03-20

**Authors:** Zhixian Ren, Ya Tan, Lili Zhao

**Affiliations:** Key Laboratory of Neuroregeneration of Jiangsu and Ministry of Education, Co-Innovation Center of Neuroregeneration, NMPA Key Laboratory for Research and Evaluation of Tissue Engineering Technology Products, Jiangsu Clinical Medicine Center of Tissue Engineering and Nerve Injury Repair, Nantong University, Nantong 226001, China; renzhixian0403@outlook.com (Z.R.); tanya2101@163.com (Y.T.)

**Keywords:** peripheral nerve system, sciatic nerve injury, axon regeneration, single-cell RNA sequence, Schwann cells

## Abstract

With the development of single-cell sequencing technology, the cellular composition of more and more tissues is being elucidated. As the whole nervous system has been extensively studied, the cellular composition of the peripheral nerve has gradually been revealed. By summarizing the current sequencing data, we compile the heterogeneities of cells that have been reported in the peripheral nerves, mainly the sciatic nerve. The cellular variability of Schwann cells, fibroblasts, immune cells, and endothelial cells during development and disease has been discussed in this review. The discovery of the architecture of peripheral nerves after injury benefits the understanding of cellular complexity in the nervous system, as well as the construction of tissue engineering nerves for nerve repair and axon regeneration.

## 1. Introduction

In mammalians, the peripheral nerve system (PNS) is responsible for the acquisition and transmission of sensations, as well as the execution of motor instructions from the central nervous system (CNS). Although neurons in PNS retain the regenerative capacity to wake the intrinsic growth programs after injury, slower regeneration and adverse conditions always lead to incomplete functional recovery and chronic pain [1,2]. To facilitate recovery, on the one hand, it is necessary to improve the regenerative ability of neurons, and a favorable regenerative environment at the regenerative end is also essential. Multiple cell types are found in peripheral nerves, including Schwann cells (SCs), fibroblasts, endothelial cells, and immune-related cells [3]. SCs were well-studied in the sciatic nerve which can wrap around axons to form the myelin sheath, speeding up the transmission of nerve signals [4]. After injury, bone marrow-derived macrophages persist for several weeks to remove debris and promote SCs’ differentiation and remyelination of regenerated axons through some macrophage-derived ligands (GAS6) [5]. In addition, survived SCs proliferate and migrate to the injury sites to build a bridge to support newborn axons [6]. Macrophages are responsible for inflammation [7], and endothelial cells participate in angiogenesis during axon regeneration [8]. The number and subtypes of these cells during axon regeneration have little been reported before single-cell RNA sequencing (scRNA-seq) application. Therefore, identifying cell heterogeneity in healthy peripheral nerves and diverse responses of subtypes after injury can help us understand the local environment changes during peripheral nerve regeneration.

Compared with the traditional detection (immunohistochemistry, in situ hybridization, electron microscopy, and transgenic mice) of cell types [9], single-cell sequencing has a great advantage in detecting cell heterogeneity and gene expression on single-cell level. Since its discovery in 2009, research based on scRNA-seq has provided a wealth of information for different fields, leading to exciting discoveries in understanding the composition and interactions of cells in human, model animals, and plants [10]. Combined with spatial omics, we can understand the composition and distribution characteristics of cell types in specific organs or tissues, as well as the molecular changes of these cell types [11]. Now, there are many scRNA-seq studies of peripheral nerves to detect cell heterogeneity during development and disease; we will first summarize the types of cells in peripheral nerves and then discuss the responses of different cell types during development and after nerve injury.

## 2. Cell Composition in Peripheral Nerve

Zeisel et al. divided cells in the nervous system into seven main categories, namely, neurons, oligodendrocytes, astrocytes, ependymal cells, peripheral glial cells (such as Schwann cells, satellite glial cells, and intestinal glial cells), immune cells, and vascular cells [12]. Cells in the peripheral nerve can be functionally divided into four categories: Schwann cells, fibroblasts, immune cells, and vascular-associated cells [13,14,15]. In adult mice, SCs and fibroblasts occupy about 65% of total detected cells, and SCs can be roughly divided into myelinating and non-myelinating SCs [13]. Fibroblasts can be divided into epineurial, perineurial, and endoneurial fibroblasts [15,16]. Immune cells have been identified including macrophages, mast cells, natural killer cells, T and B lymphocytes, and neutrophils [17,18]. Cells associated with blood vessels include endothelial cells, vascular smooth muscle cells, and peripheral cells [16,17]. There are three subtypes of endothelial cells in mouse sciatic nerves: endoneurial endothelial cells, epineurial endothelial cells, and lymphatic endothelial cells [17].

These classifications are based on the unique expression of a number of genes, which are known as cell markers. Although different studies have divided different cell subclasses according to varying combinations of cell markers, the markers of cell categories are relatively uniform. For example, *Sox10* and *Plp1* are well-known as the markers of SCs, and myelin protein genes (*Mbp*, *Mpz*, *Mag*) are used to further define myelinating SCs [17]. However, markers of non-myelinating SCs vary from study to study. Chen et al. found that *Cdh2* and *L1cam* can be used to define non-myelinated SCs [17], while Wolbert et al. suggest *Smoc2* and *Apod* are more suitable as markers of non-myelinated SCs. In adult mice, *Sfrp4* and *Pi16* performed better in the identification of epineurial fibroblasts [13]. The mainly resident immune cells in intact peripheral nerves are macrophages (8–9% of intact mouse sciatic nerve cells), myeloid cell lineage, and B and T lymphocytes; the common markers of these cell types are *Cd68*, *Lyz2*, *Cd79*, and *Cd3e*, respectively [17]. For endothelial cells in mouse sciatic nerve, *Egfl7* and *Ecscr* can be selected as their common markers [17]. To more directly show the cell type division and their related markers in the peripheral nerve, we summarized scRNA-seq studies about the peripheral nerve system as listed in Table 1 below.

## 3. Schwann Cells

### 3.1. Cell Heterogeneity of SCs during Development

Peripheral glial cells contain myelinating Schwann cells, Remak Schwann cells, repair Schwann cells, satellite glia, boundary cap-derived glia, perineurial glia, terminal Schwann cells, glia found in the skin, olfactory ensheathing cells, and enteric glia [21]. Hence, the functions of SCs are very comprehensive, including providing support and nutrition to neurons and forming myelin sheets around axons to speed up neuroelectrical signal transduction [22]. During development, neural crest stem cells are the source of Schwann cell precursors (SCPs), and the latter can further generate myelinating and non-myelinating Schwann cells. To examine the heterogeneity of the neural crest and SC lineage during development, Kastriti et al. collected more than 8000 cells from E9.5 to adult mice to perform Smartseq2 single-cell transcriptomics sequencing [23]. They found SCPs at E10.5, E11.5, and E12.5 represent a “hub” state, with markers of *Itga4*, *Serpine2*, and *Sox8*. These hub cells can later transition to iSCs by decreasing *Itga4* levels and progressive *S100β* upregulation [24] at E13.5. In this process, the Notch signal pathway promotes the generation of SCs from SCPs through *RBP-J* [25]. And then, iSCs give rise to two different types: one form neuromuscular junction SCs (tSCs) (*Cspg4*, *Itga8*, *Slitrk3*, *Cpm*, *Pou3f1*) at E18.5, and another form primed states of SCs. The latter can be further divided into two subclasses: one can continue to differentiate into myelinating or non-myelinating SCs and the other can eventually differentiate into endoneurial fibroblasts (*Igf2*, *Cd34*, *Pdgfra*) [15]. This relationship between SCs and fibroblasts had also been reported before [26,27], reminding the possibility of transformation between these two cell types.

After birth, there are six subtypes of SCs in the sciatic nerve from studies combining bulk RNA sequencing (bulk-seq), Smart-seq, and 10x genomic sequencing of sciatic nerves of mice. There are proliferating SCs (prol. SC, *Mki67*), immature SCs (iSC, *Ngfr*), pro-myelinating SCs (pmSC, *Ncmap*), myelinating SCs (mSC, *Mpz*), transition SCs (tSC, *Ncam1*), and mature non-myelinating SCs (nm(R)SC, *Ncam1*) [28]. Different from the tSCs (terminal Schwann cells) found by Kastriti et al. [21], tSCs (transition SCs) here are not neuromuscular junction SCs; they represent a transition state that only appears at postnatal day 14 (P14) and switches to mSC and nm(R)SC at P60. And prol. SC here is similar to the SCPs, showing intense proliferation and differentiation into iSCs. However, it is not clear about other differentiation potential of prol. SC. Another interesting thing is that myelinating SCs appear at an early stage at P5, while mature non-myelinating SCs do not appear until a very late stage at P60, indicating spatial and temporal specificity of the development of different subtypes of SCs. In adult mice, SCs account for nearly half (45.1%) of all sciatic nerve cells, and these SCs can also be divided into myelinating SCs (about 70%) and non-myelinating SCs (30%). Further analysis shows that myelinating SCs could be further divided into three clusters: one *Slit2* high cluster (associated with axon guidance), one *Cnp* high cluster (myelin-associated enzyme), and one cluster with moderate levels of *Cnp*, *Plp1* and *Slit2* [16]. Similar to mice, the subtypes of proliferating SCs and myelinating SCs are found in neonatal rats (P2–P4) and myelinating SCs and mature non-myelinating subtypes are found in adult rats [14].

### 3.2. Unique Repaired SC Subtype Appeared after Injury

Physical trauma to the peripheral nerves induces a series of decellularization and regenerative events related to changes in nerve structure, cell composition, and signaling. Acute damage to the nerve can lead to Wallerian degeneration, which is neurodegeneration of the distal stumps and axon debris. Schwann cells have remarkable cellular plasticity in response to nerve injury. Firstly, both myelinating SCs and non-myelinating SCs will dedifferentiate to a proliferation-activate stage. The scRNA-seq about normal and injured sciatic nerve and brachial nerve plexus revealed a distinct distribution in intact nerve, while these clusters rapidly merged into one SC cluster at 3 d after injury [17]. These merged SCs showed high expression of proliferation markers like *Mki67*, *Top2a*, *Prc1*, and *Ccna2*. Similarly, another study comparing SC subtype between development and injury also found one cluster appearing only in injured nerves. Contrasted to uninjured non-myelinating SCs, virtually all SCs from injured nerves were more similar to the neonatal SCs. This development-like transcriptional state displays upregulation of multiple growth factors, including BDNF, GDNF, and PDGFα, which has been well-studied in nerve development, nerve regeneration, and tissue repair [29,30,31]. In addition to the sciatic nerve, Renthal et al. performed single nucleus RNA-seq (snRNA-seq) on lumbar DRGs in adult mice and observed two clusters of Schwann cells in undamaged DRGs: myelinating SCs (*Mpz+*) and unmyelinating (Remak) SCs (*Mpz*−*/Scn7a*+). Repaired SCs were found at the proximal end of spinal nerve transection, and the proportion of these cells increased continuously from 2 to 7 days after spinal nerve transection [3]. Collectively, the mature myelinating and non-myelinating SCs rapidly go through dedifferentiation, forming a developmental state after nerve injury in adults, which can rapidly proliferate, differentiate, and secrete factors to participate in axon regeneration.

Additionally, the development-like transcriptional state of SCs is also identified in dermal regeneration. The deficiency of dedifferentiated SCs (dSCs) exists in diabetic wounds both in mice and humans, and increasing dSC activity through TGF-β3 can promote early wound healing [32]. Contrasted to normal skin, the number of SCs significantly increases in keloid as well as surrounding skins. These SCs are spindle-shaped, with long extensions at both ends, common with the morphology of repaired SCs, and have a positive influence on the extracellular matrix, which can promote wound healing [33,34]. These results display a development map of SCs, showing the transformation of cell subtypes of SCs during development and diseases. The subtypes of SCs are more abundant in the developing stage than those in the mature stage, and injury can lead to SCs tracing back to the immature state. This dedifferentiation state of SCs can rapidly form a suitable environment, helping axon regeneration and wound healing. In addition, in the rat model of bilateral cavernous nerve injury (BCNI), CIP2A regulates GAS6-mediated Schwann cell proliferation and is considered to be a potential target for the treatment of nerve injury after radical prostatectomy [35]; a simple artificial peripheral nerve can be simulated by placing SCs within a silicon tube, which can promote the regeneration of cavernous nerves [36]. In total, Schwann cells also play a crucial role in autonomic nerve injury in the peripheral nervous system.

## 4. Fibroblast

Fibroblasts are another important cell type in the PNS and are the most abundant cells in the endoneurial, perineural, and epineural membranes. Each nerve bundle of peripheral nerves contains endoneurial fibroblasts that are closely mixed with axons and Schwann cells. Joseph et al. found that endoneurial fibroblasts originated from the neural crest, and *Dhh+* NCPs in the nerve produced SCs and endoneurial fibroblasts in similar proportions [37]. Indeed, it had been observed years ago that SCPs with closer axons differentiated into SCs, and those with farther axons differentiated into fibroblasts [38]. Carr et al. isolated a cluster of peripheral cells (*Slc2a1/Glut1*, *Itgb4*), two clusters of epineurial fibroblasts (*Dpt*, *Pcolce2*, *Ly6c1*) and two clusters of endoneurial fibroblasts (*Osr2*, *Sox9*, *Ccl9*) from five groups of *Pdgfra*-positive mesenchymal cells in adult mice without sciatic nerve injury [15]. Besides SCs, fibroblasts are the largest cell type in the peripheral nerves [13,16]. The local fibroblasts can not only secrete type I collagen to promote SC myelination but can also guide the migration of other cells after nerve injury [39], accelerate the clearance of myelin fragments, and secrete neuro-nutrients to promote nerve injury repair [4,27].

One snRNA-seq found four clusters of fibroblasts in intact peripheral nerves in mice, which were defined into epineurial, (two clusters, *Pdgfra* and *Pcolce2*), perineurial (one cluster, *Itgb4*, *Slc2a1*) and endoneurial (one cluster, *Ccbe1*, *Adamts3*) fibroblasts [16]. However, the proportion of fibroblasts in different nerve types is diverse, and the most significant difference is that the sural nerve has twice as many epineurial fibroblast populations as other nerve types, which is due to its smaller and more sparsely distributed nerve fascicles [40]. In another study, the four clusters of fibroblasts were divided into two categories: endoneurial fibroblasts (*Sox9*, *Osr2*, *Wif1*) and epineurial fibroblasts (*Sfrp2*, *Dpt*, *Pcolce2*) in intact nerves. After the injury, a subtype of fibroblasts named differentiating fibroblasts separated from the endoneurial and epineurial fibroblasts (*Dlk1*, *Mest*, *Cilp*) [17]. The repaired fibroblast cluster with *Wif1* high expression was also detected in mouse DRG at 3 d after spinal nerve transection [3], which was consistent with the transcriptional changes of cells during tissue repair and regeneration [14].

## 5. Immune Cells

### 5.1. Macrophage

Immune cells are another important cell group in the peripheral nervous system, including macrophages, T cells, B cells, mast cells, natural killer cells (NK cells), and neutrophils. These cell types play different roles in the repair of nerve injury, with macrophages being the most active in this process [20]. According to origin, macrophages in peripheral nerves can be divided into resident macrophages and blood-derived macrophages. A number of studies have shown that resident macrophages increased rapidly and a large number of bone marrow-derived macrophages infiltrated the distal nerve after sciatic nerve injury [41,42,43]. Benefiting from the large amounts of detected cells from the intact sciatic nerve of mice, macrophages (*Aif1/Iba1*, *Cd68*) can be further divided into two sub-clusters: one cluster is epineurial macrophages (*Retnla* and *Clec10a*) and another cluster is resident macrophages [17]. In rats, bone marrow-derived macrophages infiltrated the site of nerve injury, increasing in number and peaking up to 20 times that of the control group two weeks after injury, which were found to regulate SC remyelination through *Gas6* [5]. Through immunohistochemical staining, Qian et al. found amounts of macrophages arrived at the sciatic nerve injury site within 4 d post-injury and returned to a low level at 14 d post-injury [44]. At the same time, the number of macrophages in DRG also increased rapidly, especially after spinal nerve transection [3].

In general, macrophages can be divided into the pro-inflammatory M1 type and the anti-inflammatory M2 type, and the phenotypic transformation between M1 and M2 is related to axonal damage repair [45,46]. Compared with the DRGs, the markers of macrophages displayed robust upregulation in the injured sciatic nerve stumps, with M1 macrophages activated throughout and M2 cells activated only in the later phase after nerve injury [20]. The activated M1 macrophages remove myelin debris and pathogens [47], and M2 macrophages participate in angiogenesis and stroma formation through *IL-4*, *IL-10*, and *IL-13* [48]. Cell communication analysis showed that most ligand-receptor interactions appeared at 3 d post-sciatic nerve transection in mice. At this stage of Wallerian degeneration, macrophages activated SCs and worked collaboratively with them to engulf the debris through TNFα, IL1β, CSF1, and TGFβ1 [17].

### 5.2. Other Immune Cells

Compared with scRNA-seq, snRNA-seq of sciatic nerve detects more immune-related cell types in intact sciatic nerves, including T cells (*Cd3e*, *Cd3g*), CD8+ cytotoxic T cells, CD4+ helper T cells, endoneurial macrophages (*Cx3cr1*, *Trem*), epineurial macrophages (*Clec10a*, *Cd209a*), monocytes (*Ccr2*), B cells (*Bank1*), and NK cells (*Nkg7*, *Klrb1c*) [16]. Although these immune cells cannot be distinguished in intact nerves through scRNA-seq, as the damage occurs, they proliferate, infiltrate, and thus can be better distinguished. At 3 d post-injury, macrophages, B cells, and neutrophils could be separated, and T cells, NK cells, and mast cells could only be separated at 9 d after injury [14,17]. The increase in these cells has multiple functions in tissue repair. Nadeau and Perkins et al. found neutrophils accumulated at both injury site and distal stumps after injury [49,50], and lack of neutrophils significantly reduced the clearance of nerve debris [49]. In addition, neutrophils showed strong ligand–receptor interaction of TNFα, IL-1β, CCL3, and CCL4 with macrophages; these ligands can also attract monocytes from circulating blood to the injury site [17]. The proliferation capacity of infiltrated B cells, T cells, and NK cells increases with the duration of injury, as well as cell communications [17]. Inflammatory mediators such as histamine and 5-HT were secreted by mast cells after sciatic nerve injury increasing vascular permeability, thereby recruiting more immune cell infiltration to help myelin debris removal and axon regeneration [51,52].

## 6. Endothelial Cells

Endothelial cells are an important part of blood vessels. Although different subtypes of endothelial cells are defined by different studies, they can be detected in sciatic nerves, neonatal rats, and adult mice. Only one cluster of endothelial cells (*Pecam1/cd31*, *Plvap*) was found in the sciatic nerves of newborn mice (P2–P4), but two clusters appeared in uninjured adult sciatic nerves [14,19]. Endothelial cells could also be divided into lymphatic endothelial cells (*Mmrn1*, *Prox1*, *Lyve1*) and microvascular endothelial cells (*Cldn5*) [16]. After increasing the clustering resolution, three different subtypes of endothelial cells appeared; they were epineurial endothelial cells (*Sox17*, *Spock2*, *Rgcc*), endoneurial endothelial cells (*Lrg1*, *Icam1*), and lymphatic endothelial cells (*Lyve1*, *Mmrn1*, *Flt4*) [17]. Interestingly, these three subtypes of endothelial cells merged into one cluster (*Lrg1*) in both 3 d and 9 d post-injury for the loss of their sub-cluster identity following injury [17]. The loss of markers after injury also appeared in DRG neurons, suggesting that endothelial cells may be reprogrammed to an immature state, which may be important for vascular regeneration in tissue repair [3]. Indeed, endothelial cells could not only engulf myelin debris through IgG opsonization in the spinal cord 1 week after injury but also regulate macrophage infiltration, pathologic angiogenesis, and fibrosis [53]. The neural tissue of the tubular structure arrangement network of endothelial cells shows better vascularization and axon regeneration [54]. In addition, abundant ligand-receptor interactions between endothelial cells and other cell types, like SCs, fibroblasts, and vascular smooth muscle cells have been identified [17,18], indicating endothelial cells as important participants during nerve repair and regeneration.

In conclusion, injury will induce alteration in the number and subtypes of cells in the peripheral nerve, and these proliferating or transformed cell subsets can work together to clear myelin debris, promote vascular regeneration, and build regeneration bridges, thus providing a favorable regeneration microenvironment for neurons. The schematic diagram of the distribution of cell subclasses is shown in Figure 1.

## 7. Prospect

Single-cell transcriptome sequencing, which advances the understanding of cell gene expression to the single-cell level, has shown advantages in the discovery of new cell types and the revelation of cell heterogeneity. However, current methods of cell dissociation are not very ideal, and some cell types, such as myelinating SCs, make it difficult to obtain complete individual cells. In the beginning, Yim et al. obtained a few (8–9%) myelinating SCs in mice sciatic nerves through whole-cell RNA-seq. When they changed to snRNA-seq, the detected SCs reached nearly half (45.1%) of the total cell number, and the myelinating SCs accounted for more than 30% of the total detected cells [16]. snRNA-seq is a method of obtaining the nucleus by gradient centrifugation, and then sequencing the RNA in the nucleus to obtain transcriptome information [55], unlike scRNA-seq, which utilizes the RNA of the whole cell [56]. Although snRNA-seq technology can solve the problem of cell type preference of scRNA-seq, it also loses part of the transcriptome information, the detected genes are always much smaller than that of scRNA-seq [55]. Moreover, spatial transcriptomics, which can obtain spatial location information and gene expression data of cells at the same time without cell suspension, shows a specific advantage in studying local microenvironment interaction, developmental process lineage tracking, and disease pathology. Molecular and positional changes of various cells at the site of spinal cord injury have been investigated, as well as the molecular pathology in amyotrophic lateral sclerosis through spatial transcriptomics [57,58]. Therefore, it is necessary to optimize the cell dissociation technology for peripheral nerve on the one hand, and increasing the depth and breadth by multi-omics can also give us more information on PNS, especially for axon regeneration.

Since the source of autogenous nerves is limited, a tissue-engineered nerve graft is often used to treat nerve defects in clinics. Tissue-engineered nerve grafts are artificial nerve grafts consisting of biomaterial-based scaffolds (including chitosan, fibroin protein, extracellular matrix components, polysaccharides, and metallic materials), seed cells (Schwann cells, neural stem cells) and neurotrophic factors (NGF, BDNF, GDNF, CNTF, and NTF3) [59]. Our team has been working on the research of peripheral nerve injury repair for many years, combining materials with cells, extracellular matrix, and tissue engineering to explore new methods for the treatment of peripheral nerve injury. In our previous studies about tissue-engineered nerve grafts, micro-nanofiber composite biomimetics had a significant directional guiding effect on SCs [60]; bionic peptide hydrogel scaffold promoted the transformation of M2 macrophages in situ and led to the proliferation and migration of SCs and the growth of axons [61]. We want to explore the cell heterogeneity in peripheral nerves, especially at the site of injury, which will give us more reference for screening cells to construct tissue-engineered nerves.

Besides the cell heterogeneity, cell signaling through ligand–receptor interaction can also be analyzed from the scRNA-seq [62]. In the sciatic nerve in neonatal rats, most interactions were found between endothelial cells and fibroblasts, followed by pericytes. Three days post-injury, the strong interaction transferred to epineurial fibroblasts and VSM/pericytes [17]. The ligand VEGFA binding to the receptor (EPHB2, FIT1, NRP) between fibroblasts and endothelial cells may trigger early blood vessel regeneration, which is important for axon regeneration [63]. At this stage, the interaction between macrophages and SCs was important for phagocytozing degenerated nerve fiber fragments. TNF and IL-6 from activated SCs not only activate local macrophages but also recruit monocytes to the injured site [64]. In turn, macrophages secrete pro-inflammatory factors like IL-1 which stimulate SCs to synthesize and secrete neurotrophic factors to guide axon regeneration [65]. When it comes to 9 d post-injury, the collagen–integrin, EPHRIN–EPH, and FGF–FGFR interactions between subtypes of fibroblasts were the most significant, which may respond to the recombination of the extracellular matrix during axon regeneration [66]. The spatio-temporal specific changes of cytokines and cell communication after injury can provide a reference for the addition of cytokines in tissue-engineered nerves, especially for the types of cytokines and the time of slow release.

## Figures and Tables

**Figure 1 ijms-25-03511-f001:**
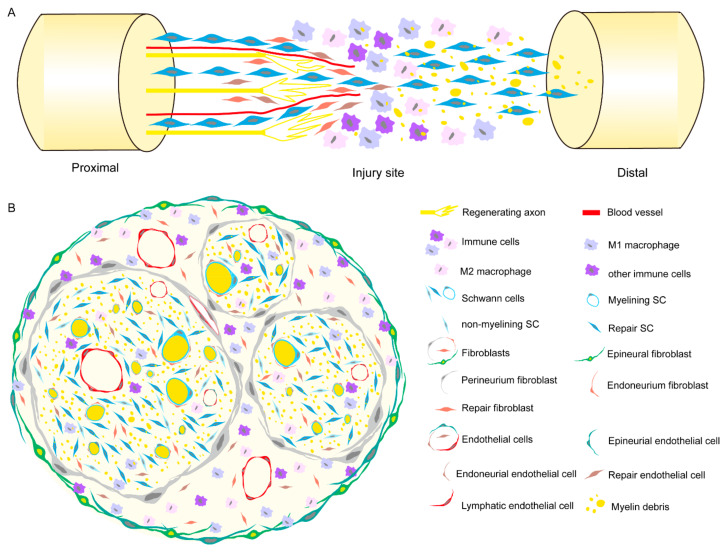
Model of cell composition at the site of axon regeneration after peripheral nerve injury. (**A**) Longitudinal section of the regeneration site of nerve. (**B**) Transverse section of the regeneration site of nerve. Repair SCs interacting with macrophages, repair fibroblasts, and repair endothelial cells to phagocytoze myelin debris, regrow microvessels, and build a bridge for guiding the sprouting axons.

**Table 1 ijms-25-03511-t001:** Cell types and related marker genes for subclasses in peripheral nerve.

Cell Type	Sample Source	Methods	Subtype and Related Markers	Reference
Schwann cell	Mouse sciatic nerves	scRNA-seq and bulk-seq (Smart-seq2 and 10X genomics	prol. SC: Mki67, Top2a nm(R)SC: *Ngfr*, *Ncam1*, *L1cam* iSC: *Ngfr*, *Ncam1*, *L1cam* tSC: *Ngfr*, *Ncam1*, *L1cam* pmSCs: *Pou3f1*, *Cdkn1c* mSCs: *Mpz*, *Mbp*, *Ncmap*	[19]
	Sciatic nerves and DRG from rat	snRNA-seq (10X genomics)	subtype 1: *Hbb*, *LOC100134871* subtype 2: *Cldn19*, *Emid1* subtype 3: *Timp3*, *Col5a3* subtype 4: *Cenpf*, *Mki67*	[18]
	Mouse sciatic nerves	snRNA-seq (10X genomics)	Schwann_M: *Prx*, *Qk*, *Mbp* Schwann_N: *Ncam1*, *Slc35f1*, *Scn7a21*, *Csmd1*	[16]
	Mouse sciatic nerves	scRNA-seq	mSCs: *Mbp*, *Mpz*, *Mag*, *Egr2* nmSCs: *Cdh2c*, *L1cam* Repair Schwann: *Mki67*, *Top2a*, *Prc1*, *Ccna2*	[17]
	Mouse DRGs	snRNA-seq	Repair Schwann: *Shh*	[3]
Fibroblast	Mouse sciatic nerves	scRNA-seq	Mesenchymal cells: *Pdgfra* Perineurial cells: *Slc2a1/Glut1*, *Itgb4*, *Msln*, *Lmo7* Epineurial cells: *Dpt*, *Pcolce2*, *Ly6c1* Endoneurial cells: *Osr2*, *Sox9*, *Ccl9*, *Wif1*, *Cdkn2a*	[15]
	Mouse sciatic nerves	snRNA-seq (10X Genomics) scRNA-seq	Epineurial fibroblast: *Pdgfra*, *Pcolce2* Perineurial fibroblast: *Itgb4*, *Slc2a1* Endoneurial fibroblast: *Ccbe1*, *Adamts3*	[16]
	Mouse sciatic nerves	scRNA-seq	Epineurial fibroblast: *Sfrp2*, *Dpt*, *Pcolce2* Endoneurial fibroblast: *Sox9*, *Osr2*, *Wif1* Differentiating fibroblast: *Dlk1*, *Mest*, *Cilp*	[17]
	Mouse DRGs	snRNA-seq	Repair fibroblasts: *Wif1*	[3]
	Mouse sciatic nerves	scRNA-seq (10X genomics)	Fibroblast: *Dpt*, *Gsn*, *Col1a1*, *Col1a2*, *Pi16*, *Sfrp4*, *Col3a1*, *Clec3b*, *Cygb*	[13]
Immune cell	Mouse sciatic nerves	snRNA-seq (10X genomics) scRNA-seq	Endoneurial macrophages: *Cx3cr1*, *Trem2* Epineurial macrophages: *Clec10a*, *Cd209a* Monocytes: *Ccr2* B cells: *Bank*1 NK cells: *Nkg7*, *Klrb1c* T cells: *Cd3g*, *Cd3e* CD8+ cytotoxic T cells: *Cd8a*, *Trac* CD4+ helper T cells: *Trac*, *Cd4*	[16]
	Mouse sciatic nerves	scRNA-seq	Epineurial macrophages: *Retnla*, *Clec10a* Macrophages: *Aif1/Iba1*, *Cd68*, *Mrc1/Cd206* Monocytes: *Ccl6*, *Fcgr3*, *Cx3cr1*, *Csf1r*, *Cd300a*, *Clec4e* Neutrophils: *S100a8*, *S100a9*, *Cxcr2*, *Cxcl2* Mast cells: *Cma1*, *Mcpt4*, *Mcpt1*, *Kit* T cells: *Cd3g*, *Cxcr6*, *Trac*, *Cd3e* NK cells: *Nkg7*, *Klrk1*, *Ncr1* B cells: *Bank1*, *Cbfa2t3*, *Taok*, *Ms4a1*, *Cd19*, *Cd79a*	[17]
	Sciatic nerves and DRG from rat	RNA-seq	M1 macrophages: *CD38*, *Grp18* M2 macrophages: *Egr2*, *Myc*, *Myc*	[20]
	Mouse sciatic nerves	scRNA-seq (10X genomics)	Epineurial Relmα Mgl1 snMacs: *Cc18*, *Cd209a*, *Cd209d*, *Fxyd2*, *Tslp*, *Mmp9* Endoneurial Relmα Mgl1 snMacs: *Ccr2*, *Cxcl1*, *ll1rl1*, *Selm*, *Pla2g2d*, *Qpct*, *Tnfsf9*	[9]
Endothelial cell	Sciatic nerves and DRG from rat	snRNA-seq (10X genomics)	Endothelial cells: *Plvap*	[18]
	Mouse sciatic nerves	scRNA-seq and Bulk-seq (Smart-seq2 and 10X genomics)	Endothelial cells-1: *Cldn5*, *Slc2a1*, *Pecam1* Endothelial cells-2: *Cd300lg*, *Pecam1*	[19]
	Mouse sciatic nerves	scRNA-seq	Endothelial cell: *Pecam1/Cd31*, *Plvap*, *Esam*	[14]
	Mouse sciatic nerves	scRNA-seq	Lymphatic endothelial cells: *Lyve1*, *Mmrn1*, *Prox1*, *Flt4* Epineurial endothelial cells: *Sox17*, *Spock2*, *Rgcc* Endoneurial endothelial cells: *Lrg1*, *Icam1*	[17]
	Mouse sciatic nerves	snRNA-seq (10X genomics) scRNA-seq	Microvascular endothelial cells: *Cldn5* Lymphatic endothelial cells: *Prox1*, *Lyve1*	[16]
Vasculature-associated smooth muscle cells (VSMCs) and pericyte	Mouse sciatic nerves	snRNA-seq (10X genomics) scRNA-seq	VSMCs: *Acta2a* Pericytes: *Pdgfrb*	[16]
	Mouse sciatic nerves	scRNA-seq	VSMCs: *Des*, *Tpm2*, *Myh11*, *Acta2*, *Mylk*, *Myom1*, *Myocd* Pericytes: *Rgs5*, *Kcnj8*, *Pdgfrb*	[17]

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
