# Peer review of "Cell Heterogeneity and Variability in Peripheral Nerve after Injury"

_ijms, 2024, doi:10.3390/ijms25063511_

Round 1

Reviewer 1 Report

Comments and Suggestions for Authors

The authors present a review article on cell heterogeneity and variability in peripheral nerve after injury. They summarize an enormous amount of complex basic science literature in an organized way. This article will serve as a roadmap for researchers in this space. This manuscript might be improved by considering the following comments and questions.

Lines 67-68: I believe there is a typo here. The authors list endoneurial endothelial cells twice.

Line 212: Please correct the citation. It currently says “Error!”

Most of the literature cited refers to somatic nerves. Would the same observations be expected in autonomic nerves of the peripheral nervous system? In urology, the most common peripheral nerve injury we deal with is the cavernosal nerve. These are commonly injured during radical prostatectomy or radiation for prostate cancer, and these patients often suffer from erectile dysfunction. Finding a way to promote regeneration of these autonomic nerves would have a major impact as prostate cancer is extremely common .

Comments on the Quality of English Language

The authors present a review article on cell heterogeneity and variability in peripheral nerve after injury. They summarize an enormous amount of complex basic science literature in an organized way. This article will serve as a roadmap for researchers in this space. This manuscript might be improved by considering the following comments and questions.

Lines 67-68: I believe there is a typo here. The authors list endoneurial endothelial cells twice.

Line 212: Please correct the citation. It currently says “Error!”

Most of the literature cited refers to somatic nerves. Would the same observations be expected in autonomic nerves of the peripheral nervous system? In urology, the most common peripheral nerve injury we deal with is the cavernosal nerve. These are commonly injured during radical prostatectomy or radiation for prostate cancer, and these patients often suffer from erectile dysfunction. Finding a way to promote regeneration of these autonomic nerves would have a major impact as prostate cancer is extremely common .

Author Response

The authors present a review article on cell heterogeneity and variability in peripheral nerve after injury. They summarize an enormous amount of complex basic science literature in an organized way. This article will serve as a roadmap for researchers in this space. This manuscript might be improved by considering the following comments and questions.

Lines 67-68: I believe there is a typo here. The authors list endoneurial endothelial cells twice.

We feel sorry for our carelessness. ln our resubmitted manuscript, the typo is revised. Thanks for your correction. We have changed one of the "endoneurial" to "epineurial", which has been marked in blue font in the revised manuscript. This is the revised sentence: There are three subtypes of endothelial cells in mouse sciatic nerves: endoneurial endothelial cells, epineurial endothelial cells, and lymphatic endothelial cells.

Line 212: Please correct the citation. It currently says “Error!”

Thanks very much for your careful checks. The mistake was caused by our negligence. We have corrected the citation in the revised manuscript.

Most of the literature cited refers to somatic nerves. Would the same observations be expected in autonomic nerves of the peripheral nervous system? In urology, the most common peripheral nerve injury we deal with is the cavernosal nerve. These are commonly injured during radical prostatectomy or radiation for prostate cancer, and these patients often suffer from erectile dysfunction. Finding a way to promote regeneration of these autonomic nerves would have a major impact as prostate cancer is extremely common.

We sincerely appreciate the valuable comments. We have checked the literature carefully and added related content to the part of 3.2. Unique repaired SCs subtype appeared after injury in the revised manuscript. What we added is: “In addition, in the rat model of bilateral cavernous nerve injury (BCNI), CIP2A regulates GAS6-mediated Schwann cell proliferation and is considered to be a potential target for the treatment of nerve injury after radical prostatectomy (PMID: 29464081); a simple artificial peripheral nerve can be simulated by placing SCs within a silicon tube, which can promote the regeneration of cavernous nerves (PMID: 27874834). In total, Schwann cells also play a crucial role in autonomic nerve injury in the peripheral nervous system.”

Reviewer 2 Report

Comments and Suggestions for Authors

The article by Ren, Z. et al. describing the cellular composition of peripheral nerve seems scientifically correct. I think it´s interesting and I only have minor comments:

- Introduction, page 1, line 1. It´s Peripheral Nervous System. The acronym is fine.

- Introduction, page 1, line 27.  This sentence is complex to read, please remove the comma before “and adverse conditions”.

- Introduction, page 1, line 29. This time, a comma is needed before “on the one hand”.

- Introduction, page 1, line 31. It´s true that axons is the common term to refer either the axons and the long dendritic expansions of sensitive neurons. However, in a general journal like this, I miss a brief mention of this point.

- Introduction, page 1, lines 34-38. The references are fine. However, the mentioned sequence is not. SCs form the myelin sheath, but after injury they frequently die due to the lack of neurotrophic support. Then, macrophagues proliferate and remove the myelin debris (endothelial cells may aid, you describe this point in section 6). Inflammation may appear, generally in the form of lymphocytes, PMNs, etc. It´s true that Schwann cells usually proliferate (in a disorderly manner) after injury. They try to recover their function, but without neurotrophic guidance they are blind. Please, refine this paragraph.

- Cell composition, page 2, line 59. I think you can remove “Except for neurons and CNS glia,”.

- Cell composition, page 2, lines 67-68. Endoneurial endothelial cells is repeated, please, check the 3 kinds of endothelial cells.

- Schwann cells, page 5, lines 89-90. This sentence is not correct. Specialized glia of ganglia and sensory receptors are not Schwann cells (they may be modified Schwann cells if you want, but not proper Schwann cells). You have a recent comprehensive review by Reed et al., in J Anat (2022, 241:1219-1234). In any case, you mention some of this special glia in your paper of 2023 in Frontiers in Neuroscience.

- Schwann cells, page 6, lines 112-115. I can´t understand this sentence.

- Schwann cells, page 6, line 153. Diabetic.

- Prospect, page 10, lines 285-288. Although these are mentioned at some parts before, you should briefly explain what is snRNA-seq and scRNA-seq before discussing this matter (you may introduce this here, but section 5.2 is also fine).

Comments on the Quality of English Language

The language is fair, with various issues. Some of them are mentioned as minor faults. There is discordance sometimes between singular and plural, and also in verbal tense. You should revise the logic of the commas. A professional revision may help.

Author Response

Comments and Suggestions for Authors

The article by Ren, Z. et al. describing the cellular composition of peripheral nerve seems scientifically correct. I think it´s interesting and I only have minor comments:

- Introduction, page 1, line 1. It´s Peripheral Nervous System. The acronym is fine.

Thank you very much for your strong support of our work.

- Introduction, page 1, line 27.  This sentence is complex to read, please remove the comma before “and adverse conditions”.

Thank you very much for your valuable suggestions. The mistake was caused by our negligence. We have removed the comma before "and disadvantage" and changed the sentence: Although neurons in PNS retain the regenerative capacity to wake the intrinsic growth programs after injury, slower regeneration and adverse conditions always lead to incomplete functional recovery and chronic pain. It is in blue in the revised manuscript.

- Introduction, page 1, line 29. This time, a comma is needed before “on the one hand”.

Thank you very much for your kind remind. We have brought "on the one hand" to the front and added a comma after it, the sentence has been revised to: on the one hand, it is necessary to improve the regenerative ability of neurons, and a favorable regenerative environment at the regenerative end is also essential. It is in blue in the revised manuscript.

- Introduction, page 1, line 31. It´s true that axons is the common term to refer either the axons and the long dendritic expansions of sensitive neurons. However, in a general journal like this, I miss a brief mention of this point.

Thank you for your advice. To reduce ambiguity, we deleted “Besides the axons,”. This is the revised sentence: Multiple cell types are found in peripheral nerves, including Schwann cells (SCs), fibroblasts, endothelial cells, and immune-related cells. We have labeled it in blue in the revised manuscript.

- Introduction, page 1, lines 34-38. The references are fine. However, the mentioned sequence is not. SCs form the myelin sheath, but after injury they frequently die due to the lack of neurotrophic support. Then, macrophagues proliferate and remove the myelin debris (endothelial cells may aid, you describe this point in section 6). Inflammation may appear, generally in the form of lymphocytes, PMNs, etc. It´s true that Schwann cells usually proliferate (in a disorderly manner) after injury. They try to recover their function, but without neurotrophic guidance they are blind. Please, refine this paragraph.

We sincerely appreciate the valuable comments. We have checked the literature carefully and added more references in the revised manuscript. We have adjusted this sequence and marked it in blue in the revised manuscript. This is the revised sentence: After injury, bone marrow-derived macrophages persist for several weeks to remove debris and promote SCs differentiation and remyelination of regenerated axons through some macrophage-derived ligands (GAS6) (PMID: 30184491). In addition, survived SCs proliferate and migrate to the injury sites to build a bridge to support newborn axons. Macrophages are responsible for inflammation, and endothelial cells participate in angiogenesis during axon regeneration.

- Cell composition, page 2, line 59. I think you can remove “Except for neurons and CNS glia,”.

Thanks for your careful checks. We have removed “Except for neurons and CNS glia,” in the revised manuscript. This is the revised sentence: Cells in the peripheral nerve can be functionally divided into four categories: Schwann cells, fibroblasts, immune cells, and vascular-associated cells.

- Cell composition, page 2, lines 67-68. Endoneurial endothelial cells is repeated, please, check the 3 kinds of endothelial cells.

We feel sorry for our carelessness. ln our resubmitted manuscript, the typo is revised. Thanks for your correction. We have changed one of the "endoneurial" to "epineurial", which has been marked in blue font in the revised manuscript. This is the revised sentence: There are three subtypes of endothelial cells in mouse sciatic nerves: endoneurial endothelial cells, epineurial endothelial cells, and lymphatic endothelial cells.

- Schwann cells, page 5, lines 89-90. This sentence is not correct. Specialized glia of ganglia and sensory receptors are not Schwann cells (they may be modified Schwann cells if you want, but not proper Schwann cells). You have a recent comprehensive review by Reed et al., in J Anat (2022, 241:1219-1234). In any case, you mention some of this special glia in your paper of 2023 in Frontiers in Neuroscience.

Thank you very much for your valuable suggestions. We have read your recommended paper, and made changes in the manuscript. This is the revised sentence: Peripheral glial cells contain myelinating Schwann cells, Remak Schwann cells, repair Schwann cells, satellite glia, boundary cap-derived glia, perineurial glia, terminal Schwann cells, glia found in the skin, olfactory ensheathing cells, and enteric glia. (PMID:34131911) Hence, the functions of SCs are very comprehensive, including providing support and nutrition to neurons and forming myelin sheets around axons to speed up neuroelectrical signal transduction.

- Schwann cells, page 6, lines 112-115. I can´t understand this sentence.

We are sorry about the unclear expression, which caused your misunderstanding. We have re-explained it in the revised manuscript and marked it with blue font. This sentence can be interpreted as follows: Different from the tSCs (terminal Schwann cells) found by Kastriti et al, tSC (transition SCs) here is not neuromuscular junction SCs, it represents a transition state that only appears at postnatal day 14 (P14) and switches to mSC and nm(R)SC at P60.

- Schwann cells, page 6, line 153. Diabetic.

Thanks for your careful checks. We are sorry for our carelessness. We have revised "diabatic" to "diabetic", which has been marked in blue font in the revised manuscript.

- Prospect, page 10, lines 285-288. Although these are mentioned at some parts before, you should briefly explain what is snRNA-seq and scRNA-seq before discussing this matter (you may introduce this here, but section 5.2 is also fine).

Thank you very much for your modification proposal. We have added some content on page 10 and marked it with blue font. This is the revised sentence: snRNA-seq is a method of obtaining the nucleus by gradient centrifugation, and then sequencing the RNA in the nucleus to obtain transcriptome information (PMID: 30586455), unlike scRNA-seq, which utilizes the RNA of the whole cell(PMID: 19349980).

Comments on the Quality of English Language

The language is fair, with various issues. Some of them are mentioned as minor faults. There is discordance sometimes between singular and plural, and also in verbal tense. You should revise the logic of the commas. A professional revision may help.

Thanks for your kind remind. We have asked for help from a native English speaker and have made the corrections in the whole manuscript. The corrected words were labeled with blue in the revised manuscript.
